# Decoder Pre-Training with only Text for Scene Text Recognition

Shuai Zhao*
School of Computer Science, Fudan University
Shanghai, China
szhao22@m.fudan.edu.cn

Yongkun Du*
School of Computer Science, Fudan University
Shanghai, China
ykdu23@m.fudan.edu.cn

Zhineng Chen†
School of Computer Science, Fudan University
Shanghai, China
zhinchen@fudan.edu.cn

Yu-Gang Jiang
School of Computer Science, Fudan University
Shanghai, China
ygj@fudan.edu.cn

## ABSTRACT

Scene text recognition (STR) pre-training methods have achieved remarkable progress, primarily relying on synthetic datasets. However, the domain gap between synthetic and real images poses a challenge in acquiring feature representations that align well with images on real scenes, thereby limiting the performance of these methods. We note that vision-language models like CLIP, pre-trained on extensive real image-text pairs, effectively align images and text in a unified embedding space, suggesting the potential to derive the representations of real images from text alone. Building upon this premise, we introduce a novel method named Decoder Pre-training with only text for STR (DPTR). DPTR treats text embeddings produced by the CLIP text encoder as pseudo visual embeddings and uses them to pre-train the decoder. An Offline Randomized Perturbation (ORP) strategy is introduced. It enriches the diversity of text embeddings by incorporating natural image embeddings extracted from the CLIP image encoder, effectively directing the decoder to acquire the potential representations of real images. In addition, we introduce a Feature Merge Unit (FMU) that guides the extracted visual embeddings focusing on the character foreground within the text image, thereby enabling the pre-trained decoder to work more efficiently and accurately. Extensive experiments across various STR decoders and language recognition tasks underscore the broad applicability and remarkable performance of DPTR, providing a novel insight for STR pre-training. Code is available at https://github.com/Topdu/OpenOCR.

## CCS CONCEPTS

• Computing methodologies → Object recognition.

## KEYWORDS

Scene text recognition, vision-language, pre-training, multi-language

*Both authors contributed equally to this research.
†Corresponding author.

ACM Reference Format:
Shuai Zhao*, Yongkun Du*, Zhineng Chen†, and Yu-Gang Jiang. 2024. Decoder Pre-Training with only Text for Scene Text Recognition. In *Proceedings of the 32nd ACM International Conference on Multimedia (MM'24), October 28-November 1, 2024, Melbourne, Australia.* ACM, New York, NY, USA, 10 pages. https://doi.org/10.1145/3664647.3681390

## 1 INTRODUCTION

Recognizing text in natural scenes, known as scene text recognition (STR), is regarded as a core task of optical character recognition (OCR). Despite significant strides in recognizing printed text images through OCR, STR encounters persistent challenges in deciphering natural text images due to complexities such as intricate background, diverse fonts and imaging conditions, etc.

To confront these challenges, many studies have been dedicated to pre-training STR models, usually employing the encoder-decoder architecture on synthetic or real text images. STR encoder pre-training methods, exemplified by CCD [24] and DiG [61], employ Masked Autoencoders [26] or contrastive learning [12] on unlabeled real text images, which drive the encoder to learn visual representations from real images, and enhancing the model's adaptability in real scenes. On the other hand, recent studies like MaskOCR [41] and TrOCR [37] train their models via two stages. For example, TrOCR is first pre-trained using hundreds of millions of printed text images, then followed by a fine-tuning on synthetic MJSynth [29] and SynthText [25] datasets. They get improved recognition results compared to the widely employed one-stage training pipeline [21, 46, 50, 64, 68, 69]. Note that both encoder and decoder are updated in these approaches. However, these approaches do not address the domain gap between synthetic and real text images. STR models trained on synthetic data, when tested on real text images, exhibit worse accuracy compared to models trained on real images [5, 17, 30], suggesting that synthetic-trained models still struggle to capture feature representations that align well with real images. The lack of large-scale labelled real text images becomes a major obstacle for building more accurate STR models. Although some progress has been achieved in English [30], this obstacle still exists for Chinese and many minority languages, which are even difficult to collect many unlabeled real images. Hence, it is imperative to explore novel STR pre-training methods that are less demanding on large-scale labelled real text images.

Recently, we observe that visual-language models like CLIP [48], trained on nearly 400 million real image-text pairs, adopt a multi-task learning approach to simultaneously optimize image and text

| Prompt Text | Real | Similarity | Synthetic | Similarity |
|---|---|---|---|---|
| a photo of a 'BMW' | 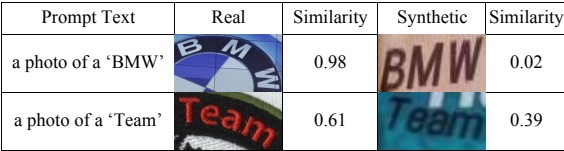 | 0.98 | | 0.02 |
| a photo of a 'Team' | | 0.61 | 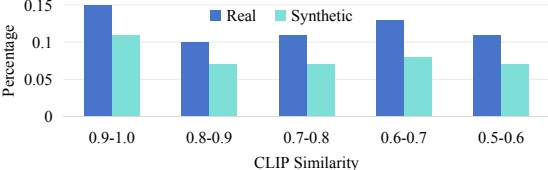 | 0.39 |

(a) CLIP similarity between distinct images and prompted labels

(b) Prompt text is more similar to real images

**Figure 1: CLIP similarity computed by cross product using the text embedding's [EOS] token and the image embedding's [CLS] token. The text embeddings are more similar to embeddings of real images rather than synthetic images.**

representations, aligning them more closely in feature space. As illustrated in Fig. 1(a), the prompt text exhibits higher similarity to real images compared to synthetic images. To further substantiate this hypothesis, we collect 49,425 samples from both SynthText [25] and real datasets (introduced in Sec. 4.1). Each sample comprises several synthetic and real text images with the same label. We compute the similarity between these images and the prompt text one-by-one following the template "a photo of a 'label'". The sum of similarities from real images serves as the real similarity of this sample and vice versa. After inspecting all the samples and images, the similarity (after Softmax) distribution is depicted in Fig. 1(b). There are 29,584 samples with a real similarity higher than 0.5, constituting 60% of the total samples. The result indicates that the CLIP text features are statistically more similar to real image features rather than synthetic image features. This suggests the feasibility of deriving potential representations of real images solely from text embeddings. In other words, performing pre-training at the decoder side by leveraging the readily available CLIP.

Building upon this premise, we introduce a novel pre-training method, named Decoder Pre-training with only text for STR (DPTR). Concretely, we utilize the CLIP text encoder to encode the prompt text, treating the resulting text embeddings as the pseudo image embeddings for decoder pre-training. However, as the text encoder is frozen, a fixed mapping relationship from the text to its embeddings is established. The lack of feature diversity may lead to overfitting of the pre-trained decoder. To mitigate this issue, we introduce an Offline Random Perturbation (ORP) strategy. This involves encoding natural images with the CLIP image encoder. The resulting image features are randomly cropped and added to the original text embeddings as background noise at a specified ratio. Subsequently, the decoder enjoys rich and diverse features for effective pre-training.

With the pre-trained decoder, we then use it to substitute the existing STR decoder, and conduct fine-tuning with synthetic or labelled real images. After this fine-tuning, the visualization of attention maps indicates that the model's attention is not chiefly directed

towards the character foreground. This phenomenon indicates that image embeddings extracted by the STR encoder contain redundant features. To remedy this issue, we introduce a Feature Merge Unit (FMU) behind the encoder. FMU employs the cross-attention mechanism to search for character features in image embeddings, and filters out redundant background features through a learnable query. This enhancement directs the model's visual attention towards the character foreground, making it easier for the decoder to decipher the character sequence.

To validate the effectiveness of DPTR, we pre-trained the decoders of three typical STR models, i.e., PARSeq [5], ABINet [21], and NRTR [50] using DPTR. The models are then applied to English, Chinese and multi-language mixed recognition tasks. All the models get improved experimental results and PARSeq reaches state-of-the-art (SOTA) accuracy. In addition, extensive ablation experiments and visualizations also verify the effectiveness of DPTR. Contributions of this paper can be summarized as follows:

- For the first time, we propose DPTR, a model-agnostic decoder pre-training method without using text images. It can be applied to many STR decoders for accuracy improvement, providing a brand-new line of insight for STR pre-training.
- We propose ORP to improve the pre-training by adding background noise to text embeddings. Meanwhile, we develop FMU that uses a learnable query to search for character foreground features and remove redundant background during fine-tuning. Both ensure the effectiveness of DPTR.
- By applying to existing STR models, DPTR achieves state-of-the-art performance on English, Chinese and multi-language mixed datasets, showcasing its remarkable performance and great universality in a wide range of STR tasks.

## 2 RELATED WORK

**Scene Text Recognition.** Scene text recognition (STR) has been extensively studied and existing methods [5, 15, 19, 21, 28, 46, 49, 58, 59, 67, 68, 70] can be classified into two categories: language-free and language-aware methods. Language-free methods predict characters directly from image features, with examples including CTC-based [22] methods like CRNN [51], SVTR [18] and Rosetta [6], ViT-based methods like ViTSTR [3], and methods that consider scene text recognition as an image classification problem [8, 29].

On the other hand, language-aware methods leverage external or internal-learned language representations to aid recognition. Methods in this category include using RNN or Transformer blocks for training semantic models. Typical examples are SRN [62] using a groundtruth-based pre-decoding, ABINet [21] refining predictions with contextual semantics via a cloze mask, NRTR [50] employing a left-to-right autoregressive decoding, and PARSeq [5] utilizing different attention masks for more nuanced semantic modeling.

**Pre-training for STR.** In order to improve the performance of STR methods, some STR pre-training studies are proposed [12, 24, 41, 61, 63]. They usually include two categories: encoder and the whole model pre-training. The encoder pre-training uses massive unlabelled real images to instruct the encoder to learn real image representations, usually through self-supervised learning such as Masked Autoencoders (MAE) [26] or contrastive learning [12]. The trained encoder can be better generalized to different

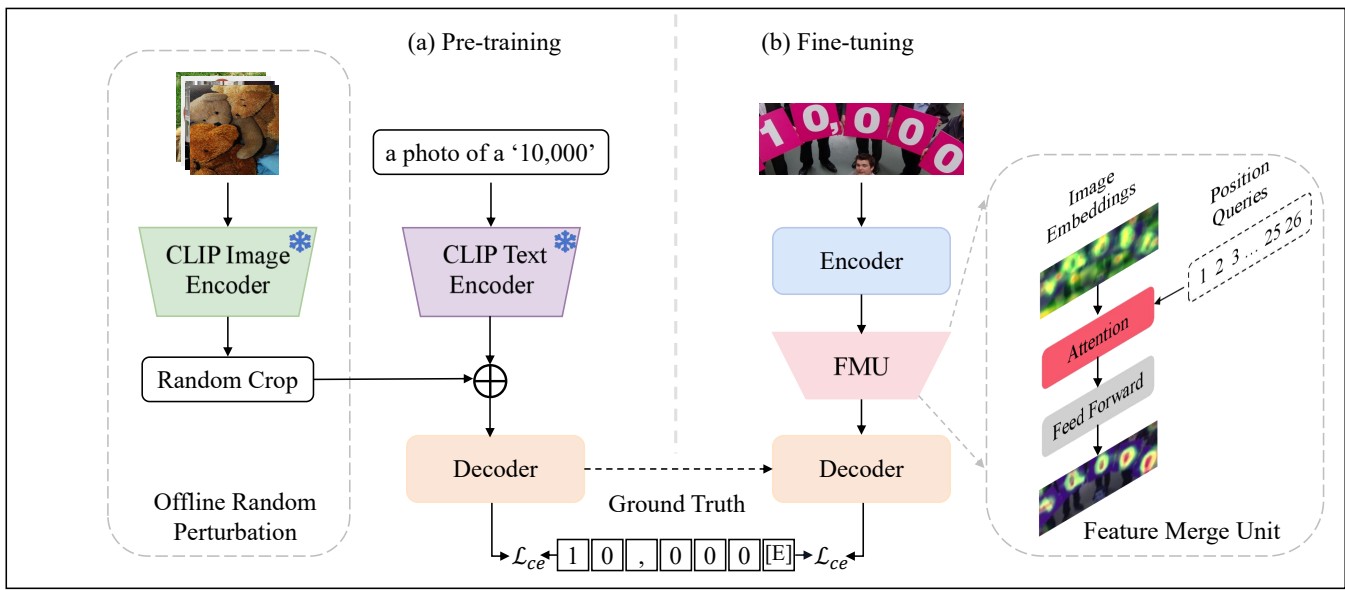

**Figure 2: The pipeline of DPTR. We pre-train the decoder by encoding the prompt text following the template "a photo of a 'label'" using the CLIP text encoder. An Offline Random Perturbation (ORP) is incorporated to prevent model overfitting. Then the entire model undergoes fine-tuning using labelled text images. A Feature Merge Unit (FMU) is developed to guide the model's visual attention towards foreground characters. $\mathcal{L}_{ce}$ denotes the cross-entropy loss.**

downstream tasks. For example, SeqCLR [1] presents a sequence-to-sequence contrastive learning framework on text images. CCD [24] introduces glyph pseudo-labels to guide the encoder focusing on the character foreground. MAERec [30] employs a ViT-based STR model and demonstrates that the model can exploit unlabelled images through a masked image modeling task.

In contrast, the whole model pre-training typically involves first pre-training a part or the whole model and then fine-tuning the whole model. For example, TrOCR [37] learns visual representations from pre-training on printed text images and fine-tuning on synthetic scene text images. Besides, it also includes BERT-style pre-training. MaskOCR [41] follows a three-stage approach including encoder pre-training, decoder pre-training, and the whole model fine-tuning. Some recent studies also evaluate synthetic data-based pre-training and real data-based fine-tuning [17, 30]. These methods mainly perform pre-training based on synthetic text images. The domain gap between synthetic and real text images remains a dominant factor restricting their performance in real scenarios. DPTR stands out from previous methods by introducing a decoder pre-training approach that does not rely on text images.

## 3 METHOD

### 3.1 Decoder Pre-training

As illustrated in the left part of Fig. 2, decoder pre-training comprises a pre-trained CLIP text encoder and a randomly initialized decoder. It aims to effectively pre-train the decoder by using prompt text. To this end, the text encoder extracts features from the prompt text. We add perturbation to these features using an Offline Random Perturbation (ORP) module. Subsequently, the decoder learns

potential representations of real images from the perturbed features and models them jointly with the prompt text.

**Text Encoder.** In English task, we adopt the text encoder of CLIP-B and use the *"a photo of a 'label'"* template to generate prompt text. The prompt text undergoes encoding into a discrete text sequence using the lower-cased byte pair encoding (BPE) [48] with a coding dictionary of size 49,152. Subsequently, the text sequence is fed into the Transformer [35] to obtain the text features. Different from CLIP, which exclusively considers features from the [EOS] token, we capture features from all the tokens. Similarly, for Chinese and multi-language mixed tasks, we employ the text encoder of Multilingual CLIP-B [9] with the same template but their own language as the label, e.g., *"a photo of a 标签'"* in Chinese. For an input text label $\hat{y}$, the text features $F_t$ can be:

$$F_t = \mathcal{T}(\hat{y}) \in \mathbb{R}^{L_t \times D} \tag{1}$$

where $\mathcal{T}(\cdot)$ denotes the CLIP text encoder, and $L_t = 78$ denotes the token length outputted by the text encoder. We directly concatenate the [EOS] token with the original 77 tokens after text projection. $D = 512$ denotes the feature dimension.

**Offline Random Perturbation (ORP).** Since the text encoder is frozen during pre-training, the obtained features are also fixed given the same prompt text, as the decoder has a fixed mapping between them. To resolve this problem, we randomly encode 10,000 natural images from COCO2017 [38] dataset using the CLIP image encoder. Subsequently, we randomly select the features of one image, and add them as background noise to the text features. The obtained image features are saved locally for facilitating the subsequent pre-training. Through this straightforward implantation, different text features can be obtained given the same prompt text, thus largely

enriching the diversity of features and effectively preventing model overfitting. The perturbed features $F_p$ can be written as:

$$F_p = F_t + \lambda \cdot C(\mathcal{I}(\tilde{x})) \in \mathbb{R}^{L_t \times D} \tag{2}$$

where $\tilde{x}$ is the randomly selected natural image, $\mathcal{I}(\cdot)$ denotes the CLIP image encoder, $C(\cdot)$ denotes the crop strategy that randomly selects $L_t$ tokens from the CLIP image features, and $\lambda$ is a hyperparameter controlling the weight of background noise.

**Decoder.** For decoder pre-training, typically, the objective is to enable the decoder to search for potential real image representations from perturbed features $F_p$, integrate them with contextual information, and facilitate text recognition ultimately. To this end, we have chosen three language-aware STR models, ABINet [21], NRTR [50] and PARSeq [5], for text recognition. Their decoders are different thereby evaluations on them can reveal the universality of our pre-training. For the input text $\hat{y}$, the decoder predictions $y_m$ can be uniformly expressed as:

$$y_m = Dec(F_p, \hat{y}, m) \in \mathbb{R}^{(T+1) \times (S+1)} \tag{3}$$

where $Dec(\cdot)$ denotes the decoder, $m$ is an attention mask. It is a permutation-derived autoregressive (AR) mask dor PARSeq, a fixed left-to-right causal mask for NRTR, and a cloze mask for ABINet. $T$ denotes the text length, and $T + 1$ is because the [BOS] token is added to the text. $S$ is the size of character set, and $S + 1$ is because we use [EOS] to mark the end of the sequence.

**Loss Function.** For the given text label $\hat{y}$ and the prediction $y_m$, the loss function can be uniformly expressed as:

$$\mathcal{L} = \mathcal{L}_{dec}(y_{m_i}, \hat{y}) \tag{4}$$

where $\mathcal{L}_{dec}(\cdot)$ denotes the decoder loss function. It is the arithmetic mean of the cross-entropy losses obtained from the K-attention masks for PARSeq, the cross-entropy loss of $y_m$ and $\hat{y}$ for NRTR, and the weighted average of the three losses in [21] for ABINet.

## 3.2 Model Fine-tuning

With the pre-trained decoder, we then employ a fine-tuning stage as illustrated in Fig. 2 to improve the performance of existing STR models. The model comprises a randomly initialized encoder, a randomly initialized feature merge unit (FMU), and a pre-trained decoder. The image encoder extracts visual features from the input image, which are processed by FMU and then fed into the pre-trained decoder for joint semantic modeling.

**Visual Encoder.** For an input image $X \in \mathbb{R}^{W \times H}$ and the patch size $(P_w, P_h)$, the image features $F_i$ can be represented as:

$$F_i = Enc(X) \in \mathbb{R}^{\frac{WH}{P_w P_h} \times D} \tag{5}$$

where $Enc(\cdot)$ denotes the encoder, where ABINet employs ResNet [27] and Transformer units [62], while PARSeq and NRTR utilize Vision Transformer (ViT) [16].

**Feature Merge Unit (FMU).** FMU serves as an adapter to transfer the features extracted by the STR encoder to features that are more compatible with the pre-trained decoder. We first directly fine-tune the whole model without FMU. When converged, we visualize attention maps on image features $F_i$, it is observed that the encoder does not focus on foreground characters (see Sec. 4.3), indicating that redundant features are included. To address this issue, we introduced an FMU behind the image encoder. The FMU

employs the cross-attention mechanism to select $F_i$ features focusing on character foreground through a learnable query $q$, and the resulting condensed features $F_u$ can be represented as:

$$F_u = MHA(F_i, q) + FFN \in \mathbb{R}^{L_u \times D} \tag{6}$$

where $MHA(\cdot)$ denotes the Multi-head Attention, $FFN$ is the feed forward network, and $L_u$ is a hyperparameter that controls the number of tokens outputted by FMU. By employing the cross-attention mechanism above, FMU adaptively selects features that can enhance the recognition accuracy. Note that smaller $L_u$ means a more condensed representation of visual features.

**Decoder.** As shown in Fig. 2, we start fine-tuning by using exactly the pre-trained decoder, which is the same as the existing STR decoders in architecture. For instance, in case of PARSeq, the decoder includes the decoding layer, head, text embedding, and position query. During fine-tuning, the pre-trained decoder updates parameters according to the fused image features $F_u$ and contextual features. The prediction $y_m$ can be formulated as:

$$y_m = Dec(F_u, \hat{y}, m) \in \mathbb{R}^{(T+1) \times (S+1)} \tag{7}$$

Note that the decoder employs a cross-attention-based decoding scheme, where the text features are the query and $F_u$ is the key and value. The text features are extracted following their STR methods.

**Loss Function.** The loss function for fine-tuning is the same as that of the pre-training stage and is omitted here.

## 4 EXPERIMENT

### 4.1 Datasets

**Pre-training dataset.** To facilitate a fair comparison with existing methods, we generate text prompts by extracting labels from the synthetic datasets MJSynth (MJ) [29] and SynthText (ST) [25]. After de-duplication, we obtain approximately 380,000 English labels for pre-training on English. Similarly, we extract labels from the Chinese text recognition benchmark (BCTR) [11] and acquire around 700,000 Chinese labels for pre-training on Chinese. For multi-language mixed task, we obtain 150,000 labels from the synthetic datasets SynthMLT [7], which encompasses 9 languages including Chinese, Japanese, Korean, Bangla, Arabic, Italian, English, French, and German. The labels are encoded to text embeddings using the CLIP text encoder.

**Fine-tuning dataset.** Similar to prior research [5, 63], for English task, we utilize MJ and ST as the synthetic data. The two datasets have approximately 17 million synthetic text images in total. The real data employed include COCO-Text (COCO) [56], RCTW [53], Uber-Text (Uber) [66], ArT [14], LSVT [55], MLT19 [43], ReCTS [65], TextOCR [54], and Open Images [33] annotations from the OpenVINO toolkit [34], encompassing around 3 million text images depicting real scenes. For Chinese task, we adopt BCTR as the dataset, which aggregates four types of Chinese text recognition subsets: Scene, Web, Document, and Handwriting. The dataset contains about 1 million Chinese text images in total. For multi-language mixed task, we adopt MLT17 [44] and MLT19 [43] as the datasets. They together contain about 150,000 text images, covering 10 languages including Arabic, Bengali, Chinese, Devanagari, English, French, German, Italian, Japanese, and Korean.

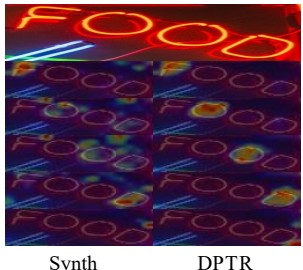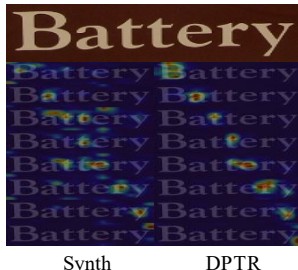

Synth          DPTR          Synth          DPTR

**Figure 3: Two examples of decoder attention map comparison between *Synth* (left) and *DPTR* (right).**

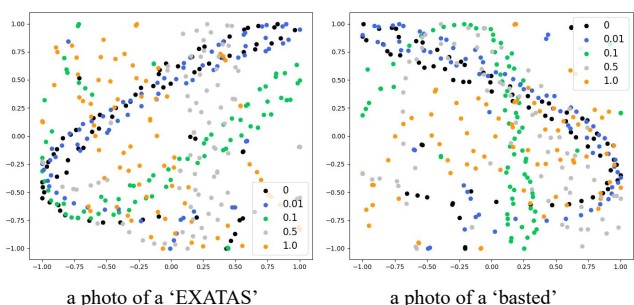

a photo of a 'EXATAS'          a photo of a 'basted'

**Figure 4: Comparison of CLIP text feature distribution with different noise ratios. Each is represented by a distinct color.**

**Table 1: Comparison between *Base*, *Synth* and *DPTR*. *freeze* denotes freezing the decoder during fine-tuning.**

| Method | IIIT5K | SVT | IC13 | IC15 | SVTP | CUTE | Avg |
|---|---|---|---|---|---|---|---|
| Base | 98.5 | 97.7 | 97.4 | 89.1 | 94.1 | 95.8 | 95.5 |
| Synth$_{freeze}$ | 98.6 | 97.7 | 97.3 | 89.2 | 92.7 | 95.8 | 95.2 |
| Synth | 98.8 | 97.2 | 98.0 | 90.3 | 93.2 | 96.9 | 95.8 |
| DPTR$_{freeze}$ | 98.5 | 98.1 | 97.1 | 89.2 | 94.6 | 96.9 | 95.7 |
| DPTR | 98.9 | 97.9 | 98.4 | 89.7 | 95.8 | 97.9 | **96.4** |

**Table 2: Ablation study on pre-training with different ORP noise ratios. 0 denotes pre-training without ORP.**

| $\lambda$ | IIIT5K | SVT | IC13 | IC15 | SVTP | CUTE | Avg |
|---|---|---|---|---|---|---|---|
| 0 | 98.9 | 97.9 | 98.4 | 89.7 | 95.8 | 97.9 | 96.4 |
| 0.01 | 98.7 | 98.2 | 98.5 | 90.5 | 94.7 | 98.6 | 96.5 |
| 0.1 | 98.9 | 98.5 | 98.3 | 90.5 | 96.3 | 99.0 | **96.9** |
| 0.5 | 98.9 | 98.0 | 98.1 | 90.2 | 94.7 | 96.2 | 96.0 |
| 1 | 98.8 | 98.3 | 98.4 | 90.1 | 94.1 | 96.9 | 96.1 |

**Test benchmark.** We recruit the following test sets for English task: IIIT 5k-word (IIIT5k) [42], CUTE80 (CUTE) [2], Street View Text (SVT) [57], SVT-Perspective (SVTP) [45], ICDAR 2013 (IC13) [32], and ICDAR 2015 (IC15) [31].

For Chinese task, we utilize the test sets of BCTR, which are also further categorized into four subsets: Scene, Web, Document, and Handwriting. For multi-language mixed task, we use the validation set of MLT17 for test only due to the unavailability of MLT19 test data. This set encompasses 6 subsets covering 9 languages: Chinese, Japanese, Korean, Bangla, Arabic, and Latin (Italian, English, French, and German).

## 4.2 Experimental Settings

The input image is resized to $32 \times 128$ for both English and multi-language mixed tasks. For Chinese task, we resize the input image to $32 \times 256$. The patch size is set to $4 \times 8$ for all languages. The maximum text length is restricted to 25 characters. We pre-train the model on 2 NVIDIA RTX A6000 GPUs with a batch size of 512, and then fine-tune it with a batch size of 384. Hyperparameters include an initial learning rate of 7e-4 without weight decay.

## 4.3 Ablation Study

We conduct ablations to verify the effectiveness of the proposed decoder pre-training, ORP and FMU. For brevity, ***Synth*** denotes the method pre-trained with synthetic images.

**The effectiveness of decoder pre-training.** We conduct a comparative experiment with *Base*, *Synth*, and *DPTR* sharing the same model structure and experimental setup. The primary distinction of the three methods lies in: *Base* trains directly on the real datasets without pre-training, *Synth* undergoes pre-training with synthetic images before fine-tuning on real images, and *DPTR* pre-trains with

text only before fine-tuning on real images. Experimental results presented in Tab. 1 show that *DPTR* improves average accuracy by 0.9% compared to *Base* and by 0.6% over *Synth*. Furthermore, when the pre-trained decoder is frozen during fine-tuning, the model experiences only a marginal accuracy decrease of 0.7%, indicating the effectiveness of the text pre-trained decoder.

As shown in Fig. 3, we compare the pre-trained decoder attention maps between *Synth* and *DPTR*. The results reveal that *Synth* exhibits more pronounced attention drift, suggesting a higher susceptibility to interference from intricate backgrounds in real images. In contrast, attention of *DPTR* is mostly located on the corresponding characters, indicating a more accurate alignment between image embeddings and text embeddings. We also visualize the character distribution of *Synth* and *DPTR*, as shown in Fig. 5, where each circle is a character and its color represents the character category. The symbol '+' denotes 'n' in the image labelled 'nVIDIA', while 'x' represents 't' in the image labelled 'tO'. Due to overlapping with the background, *Synth* incorrectly predicts 'n' as '2'. Similarly, character 't' is obscured such that *Synth* misses it. Meanwhile, it incorrectly identifies 'O' as '0'. In contrast, for *DPTR* the two misidentified characters fall into the correct character categories. Since *Synth* and *DPTR* only differ in the pre-training step, the results in Tab. 1, Fig. 3 and Fig. 5 clearly indicate the effectiveness of the proposed decoder pre-training.

**The effectiveness of ORP.** To investigate the impact of adding random perturbation to decoder pre-training, we ablate the noise ratio that balances the weight of text features and randomly selected visual features. The results are presented in Tab. 2. The model's accuracy first experiences an increase when the noise ratio is small, and a decrease when the noise ratio further goes up. This phenomenon is attributed to that excessive noise alters the text features

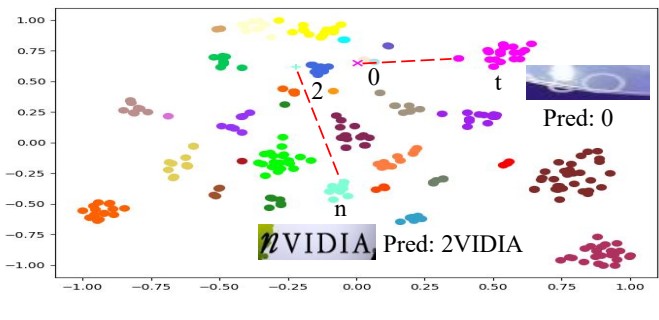
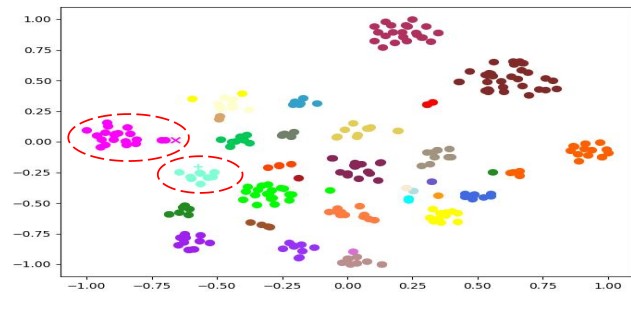

(a) Pre-trained with synthetic images

(b) Pre-trained with text by DPTR

**Figure 5: Character distribution visualization of the decoder pre-trained by *Synth* and *DPTR*. Point color represents the character category. In (a), '+' and 'x' represent two incorrect predictions, e.g., '2' and '0', whereas in (b), they are correctly recognized.**

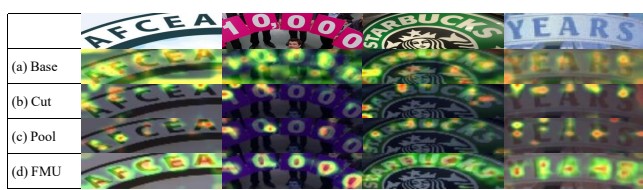

**Figure 6: Attention maps of different feature fusing methods.**

**Table 3: Comparison of different feature fusing methods.**

| Method | IIIT5K | SVT | IC13 | IC15 | SVTP | CUTE | Avg |
|---|---|---|---|---|---|---|---|
| Cut | 98.1 | 96.9 | 97.7 | 89.3 | 91.8 | 95.5 | 94.9 |
| Pool | 98.0 | 96.9 | 97.2 | 88.9 | 93.2 | 95.8 | 95.0 |
| FMU | 99.5 | 99.2 | 98.5 | 91.8 | 97.1 | 98.6 | **97.5** |

too much, leading the decoder to acquire incorrect representations. In contrast, introducing a small ratio of noise effectively prevents model overfitting, meanwhile without significantly altering the distribution of text features. As depicted in Fig. 4, the distributions of text features are significantly altered when setting $\lambda = 0.5$ or $\lambda = 1$, which display quite different shapes compared to the raw distribution ($\lambda = 0$). Conversely, for $\lambda = 0.01$, the added noise is minor, resulting in little deviation from the raw distribution. For $\lambda = 0.1$, although the distribution changes a lot from the raw, it still holds a certain geometric shape. This analysis vividly illustrates that introducing a small noise ratio through ORP enriches the diversity of pre-training features, thereby improving the performance.

**The effectiveness of FMU.** It is not surprising that the features from the CLIP text encoder are different from those extracted by the image encoder employed in fine-tuning. FMU serves as an adapter to bridge this gap. To gain insights into the role of FMU, we first omit the FMU module and fine-tune the model. Visualizations of the last self-attention layer of the image encoder are shown in Fig. 6(a), it is apparent that the attention is not solely focused on the character foreground. Instead, a portion of attention is directed towards the background. This observation suggests that not all the extracted features are useful and positively contribute to the recognition. Some of them are redundant and may hinder the recognition.

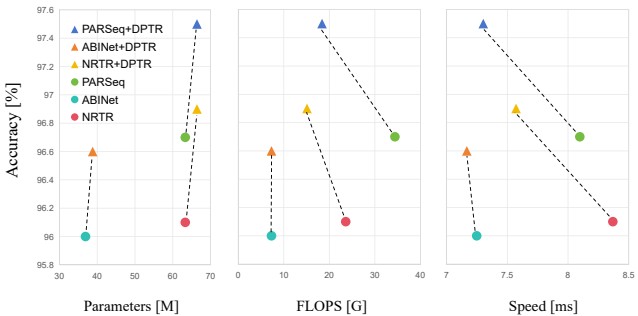

**Figure 7: Accuracy-parameter/computational cost/inference speed plots of PARSeq, ABINet and NRTR. *+DPTR* means combined with our DPTR.**

**Table 4: Ablation study on the size of outputted features in FMU during fine-tuning. *w/o* denotes without FMU.**

| $L_u$ | IIIT5K | SVT | IC13 | IC15 | SVTP | CUTE | Avg |
|---|---|---|---|---|---|---|---|
| w/o | 98.9 | 98.5 | 98.3 | 90.5 | 96.3 | 99.0 | 96.9 |
| 20 | 99.2 | 98.9 | 98.1 | 90.8 | 95.8 | 97.9 | 96.8 |
| 26 | 99.5 | 99.2 | 98.5 | 91.8 | 97.1 | 98.6 | **97.5** |
| 30 | 99.3 | 98.6 | 98.2 | 90.9 | 96.4 | 98.3 | 97.0 |
| 40 | 99.1 | 98.3 | 98.4 | 90.9 | 95.8 | 98.6 | 96.9 |

With this observation, we propose FMU to mitigate the feature redundancy and address the issue of attention not focusing. As depicted in Fig. 6(d), when FMU is equipped, hotspots of the attention maps are mostly concentrated on character foreground, which vividly indicates that FMU successfully extracts useful image features and discards redundant ones, explaining why accuracy gains are obtained. In addition to using the proposed FMU to distinct features, there also are other ways to fuse these features. We ablate two typical of them. The first is *Cut* that directly truncates the first $L_u$ tokens from the image encoder. While the second is *Pool* that denotes pooling all the tokens into $L_u$ tokens using *AdaptiveAvg-Pool1d*. Their comparing results ($L_u$=26) are given in Tab. 3. *Cut* exhibits the worst performance and *Pool* also reports an accuracy

**Table 5: Accuracy comparison with existing methods across six English benchmarks.** *Avg* **represents the arithmetic average of IIIT5K, SVT, IC13 (857), IC15 (1811), SVTP, and CUTE.**

| Method | IIIT5K 3000 | SVT 647 | IC13 857 | IC13 1015 | IC15 1811 | IC15 2077 | SVTP 645 | CUTE 288 | Avg | IIIT5K 3000 | SVT 647 | IC13 857 | IC13 1015 | IC15 1811 | IC15 2077 | SVTP 645 | CUTE 288 | Avg |
|---|---|---|---|---|---|---|---|---|---|---|---|---|---|---|---|---|---|---|
| Training data | Synthetic Datasets | | | | | | | | | Real Datasets | | | | | | | | |
| TRBA[4] | 96.3 | 92.8 | 96.3 | 95.0 | 84.3 | 80.6 | 86.9 | 91.3 | 91.3 | 98.6 | 97.0 | 97.6 | 97.6 | 89.8 | 88.7 | 93.7 | 97.7 | 95.7 |
| ViTSTR-S[3] | 94.0 | 91.7 | 95.1 | 94.2 | 82.7 | 78.7 | 83.9 | 88.2 | 89.3 | 98.1 | 95.8 | 97.6 | 97.7 | 88.4 | 87.1 | 91.4 | 96.1 | 94.6 |
| SVTR-B[18] | 96.0 | 91.5 | 97.1 | - | 85.2 | - | 89.9 | 91.7 | 91.9 | 97.7 | 95.8 | 95.2 | 95.7 | 89.4 | 88.5 | 91.8 | 96.2 | 94.3 |
| MaskOCR-B[41] | 95.8 | 94.7 | **98.1** | - | 87.3 | - | 89.9 | 89.2 | 92.5 | - | - | - | - | - | - | - | - | - |
| TrOCR-B[37] | 90.7 | 91.0 | 97.3 | 96.3 | 81.1 | 75.0 | 90.7 | 86.8 | 89.6 | - | - | - | - | - | - | - | - | - |
| CCD-B [24] | 97.2 | 94.4 | - | 97.0 | 87.6 | - | 91.8 | 93.3 | - | 98.0 | 97.8 | - | 98.3 | 91.6 | - | 96.1 | 98.3 | - |
| LPV-B[64] | 97.3 | 94.6 | 97.6 | 96.8 | 87.5 | **85.2** | 90.9 | **94.8** | 93.8 | 97.6 | 98.8 | 97.8 | 97.9 | 89.1 | 88.1 | 94.0 | 97.2 | 95.8 |
| MGP-STR[15] | 96.2 | 94.9 | 97.6 | 96.6 | **87.9** | 83.8 | 90.2 | 89.2 | 92.7 | 98.4 | 98.3 | **98.6** | 98.4 | 91.1 | 89.8 | 96.6 | 97.9 | 96.8 |
| CDistNet[69] | 96.4 | 93.5 | 97.4 | 95.6 | 86.0 | 82.5 | 88.7 | 93.4 | 92.6 | 97.9 | 95.4 | 96.6 | 96.5 | 89.1 | 88.0 | 92.7 | 97.2 | 94.8 |
| ABINet++[20] | 96.2 | 93.5 | 97.4 | 95.7 | 86.0 | 85.1 | 89.3 | 89.2 | 91.9 | 97.1 | 96.1 | 98.1 | 97.1 | 89.2 | 86.0 | 92.2 | 94.4 | 94.0 |
| SIGA[23] | 96.6 | 95.1 | 97.8 | 96.8 | 86.6 | 83.0 | 90.5 | 93.1 | 93.3 | - | - | - | - | - | - | - | - | - |
| LISTER[13] | 96.8 | 93.5 | 97.7 | **97.3** | 87.2 | 83.5 | 89.5 | 89.6 | 92.4 | 98.4 | 98.5 | **98.6** | **98.6** | 89.7 | 87.5 | 94.0 | 94.8 | 95.7 |
| OTE-B [60] | 96.4 | 95.5 | 97.9 | - | 86.8 | - | **91.9** | 90.3 | 93.1 | - | - | - | - | - | - | - | - | - |
| NRTR [50] | 95.6 | 92.9 | 96.6 | 95.0 | 84.1 | 80.5 | 86.4 | 88.5 | 90.7 | 99.0 | 97.2 | 98.0 | 98.0 | 90.3 | 89.3 | 94.6 | 97.6 | 96.1 |
| ABINet [21] | 95.3 | 93.4 | 97.1 | 95.0 | 83.1 | 79.1 | 87.1 | 89.7 | 91.0 | 98.6 | 97.8 | 98.0 | 97.8 | 90.2 | 88.5 | 93.9 | 97.7 | 96.0 |
| PARSeq [5] | 97.0 | 93.6 | 97.0 | 96.2 | 86.5 | 82.9 | 88.9 | 92.2 | 92.5 | 99.1 | 97.9 | 98.3 | 98.4 | 90.7 | 89.6 | 95.7 | 98.3 | 96.7 |
| NRTR+DPTR | 95.2 | 93.7 | 97.2 | 95.8 | 85.4 | 81.3 | 88.4 | 91.6 | 91.9 | 99.2 | 97.8 | 98.1 | 98.1 | **91.8** | 90.6 | 95.7 | **98.6** | 96.9 |
| ABINet+DPTR | 95.9 | 94.6 | 96.7 | 95.3 | 85.4 | 80.9 | 87.9 | 90.6 | 91.9 | 98.7 | 98.5 | 97.9 | 97.6 | 91.3 | 89.2 | 94.9 | 98.3 | 96.6 |
| PARSeq+DPTR | **97.6** | **96.0** | 97.9 | 97.0 | 87.2 | 83.7 | **91.9** | 94.1 | **94.1** | **99.5** | **99.2** | 98.5 | 98.4 | **91.8** | **90.8** | 97.1 | **98.6** | **97.5** |

decrease of 2.5% compared to *DPTR*. We attribute these discrepancies to the loss of vital visual information by using *Cut* and *Pool*. To confirm this, We visualize the attention maps for both methods in Fig. 6(b) and (c). It is evident that both methods exhibit attention deficits, which leads to degraded performance. In contrast, the proposed FMU keeps sufficient attention on the character foreground, thereby yielding the best performance. The result demonstrates that the cross-attention-based feature fusion can retain the vast majority of useful features extracted from the image encoder.

Furthermore, we conduct a comparative experiment on the size of the outputted feature in FMU. Larger size means more features are retained and vice versa. As depicted in Tab. 4, the model achieves the best performance when $L_u = 26$. This result can be attributed to that we set the maximum character length for the text to 25. With the addition of [BOS], the decoder can generate up to 26 tokens. Consequently, by setting $L_u = 26$ in FMU, an implicit mapping between FMU tokens and characters can be established directly. In contrast, setting $L_u$ less or greater than 26 may result in complicated token-character mapping, leading to the features being less effectively utilized. According to our experimental setting, the image encoder will output 128 tokens for English and multi-language mixed tasks, and 256 tokens for Chinese. Setting $L_u = 26$ means only a small portion of tokens are preserved. On one hand, it implies the image features are indeed redundant. On the other hand, it also means that the decoder can be computed more efficiently. As depicted in Fig. 7, while adding FMU marginally increases the model parameters, the recognition accuracy is improved for all the three STR models. Meanwhile, adding DPTR the computational

cost becomes lower, and the inference speed is faster especially for those autoregressive-based models. The results above convincingly verify that FMU can lead to more accurate and efficient STR.

## 4.4 Comparisons with State-of-the-Arts

We conduct extensive comparisons with existing STR models on English, Chinese, and multilingual tasks. +*DPTR* denotes that the method is combined with our DPTR.

**English Benchmarks.** In Tab. 5, we give the results of three STR models (i.e., NRTR, ABINet, and PARSeq) combined with DPTR and sixteen existing models. Taking *PARSeq* + *DPTR* as an example, the accuracy on synthetic datasets increased by 0.3% compared to LPV-B, the best previous model, and on real datasets the improvement against MGP-STR, also the best previous model, is 0.7%. Meanwhile, the models trained on synthetic and real data increase the accuracy by 1.6% and 0.8%, respectively, compared to the raw PARSeq. Similar improvements are also observed when *NRTR* + *DPTR v.s.* NRTR, and *ABINet* + *DPTR v.s.* ABINet. These results show the merits of incorporating DPTR.

When inspecting the pre-training related models, *PARSeq* + *DPTR* trained on synthetic data gains accuracy improvements of 1.6% and 4.5% compared to MaskOCR-B and TrOCR-B, respectively. The most substantial improvements are observed on SVT and CUTE, where accuracy increases are 1.3% and 5.0% on SVT, and 4.9% and 7.3% on CUTE. These results suggest that DPTR excels in handling street text and curved text images, which are typical difficulties for most existing models. These remarkable improvements clearly indicate the superiority of DPTR as a STR pre-training technique.

                                                      Shuai Zhao, Yongkun Du, Zhineng Chen, and Yu-Gang Jiang

**Table 6: Comparison on challenging English datasets.**

| Method | ArT 35,149 | COCO 9,825 | Uber 80,551 | ArT 35,149 | COCO 9,825 | Uber 80,551 |
|---|---|---|---|---|---|---|
| Training data | Synthetic Datasets | | | Real Datasets | | |
| CRNN[51] | 57.3 | 49.3 | 33.1 | 66.8 | 62.2 | 51.0 |
| ViTSTR-S[3] | 66.1 | 56.4 | 37.6 | 81.1 | 74.1 | 78.2 |
| TRBA[4] | 68.2 | 61.4 | 38.0 | 82.5 | 77.5 | 81.2 |
| LISTER-B[13] | 70.1 | 65.8 | **49.0** | 79.6 | 75.1 | 75.6 |
| OTE [60] | 69.1 | 64.5 | 47.8 | - | - | - |
| NRTR[50] | 67.0 | 60.7 | 39.6 | 83.7 | 78.5 | 84.8 |
| ABINet[21] | 65.4 | 57.1 | 34.9 | 81.2 | 76.4 | 71.5 |
| PARSeq[5] | 70.7 | 64.0 | 42.0 | 84.5 | 79.8 | 84.5 |
| NRTR+DPTR | 68.0 | 64.0 | 40.9 | 84.1 | 79.9 | 84.7 |
| ABINet+DPTR | 69.0 | 66.6 | 42.5 | 81.3 | 76.2 | 74.5 |
| PARseq+DPTR | **72.4** | **69.0** | 43.3 | **85.0** | **81.3** | **87.7** |

**Table 7: Comparison on four standard Chinese datasets.**

| Method | Doc | Hand | Scene | Web | Avg |
|---|---|---|---|---|---|
| SAR[36] | 93.8 | 31.4 | 62.5 | 54.3 | 60.5 |
| MORAN[40] | 95.8 | 39.7 | 51.8 | 49.9 | 59.3 |
| ASTER[52] | 93.1 | 38.9 | 54.5 | 52.3 | 59.7 |
| SEED[47] | 93.7 | 32.1 | 49.6 | 46.3 | 55.4 |
| SRN[62] | 96.7 | 18.0 | 60.1 | 52.3 | 56.8 |
| TransOCR[10] | 97.1 | 53.0 | 71.3 | 64.8 | 71.6 |
| MASTER[39] | 84.4 | 26.9 | 62.8 | 52.1 | 56.6 |
| MaskOCR-B[41] | 99.3 | 63.7 | 73.9 | 74.8 | 77.9 |
| MaskOCR-L[41] | **99.4** | **67.9** | 76.2 | 76.8 | 80.1 |
| SVTR-B[18] | 98.2 | 52.2 | 71.7 | 73.8 | 74.0 |
| CCR[63] | 98.3 | 60.3 | 71.3 | 69.2 | 74.8 |
| NRTR[50] | 97.5 | 55.7 | 68.5 | 68.8 | 72.6 |
| ABINet[21] | 98.2 | 53.1 | 66.6 | 63.2 | 70.3 |
| PARSeq[5] | 96.9 | 55.9 | 74.3 | 74.6 | 75.4 |
| NRTR+DPTR | 97.5 | 56.4 | 75.2 | 75.8 | 76.2 |
| ABINet+DPTR | 95.8 | 51.0 | 70.7 | 72.8 | 72.6 |
| PARSeq+DPTR | 98.9 | 64.4 | **80.0** | 79.6 | **80.7** |

To further validate the performance of DPTR, we conduct evaluations on Art, COCO, and Uber datasets, which are typically more challenging compared to previous benchmarks. As depicted in Tab. 6, compared to PAPSeq, *PARSeq + DPTR* trained on synthetic data achieves accuracy improvements of 1.7%, 5.0%, and 1.3%, and when trained on real datasets, the improvements are 0.5%, 1.5%, and 3.2%, respectively. It achieves the best accuracy among five of the six comparisons. Similarly, *NRTR + DPTR* and *ABINet + DPTR* both report improvements compared to their raw counterparts.

**Chinese Benchmarks.** We also train a DPTR for Chinese through Multilingual CLIP and evaluate it on BCTR. As depicted in Tab. 7, compared with MaskOCR-L, the previous SOTA method *PARSeq + DPTR* reports an average accuracy of 80.7%. It is also the new state of the art. *PARSeq + DPTR* gains accuracy improvements of 3.8% and 2.8% on *Scene* and *Web*, respectively. However, it is worse than MaskOCR-L on *Doc* and *Hand*. This is because *Doc* is synthesized using a text rendering tool, and the Handwriting style in *Hand* is more similar to synthetic text images. Both are less similar to real scene text. The observation indicates that *PARSeq + DPTR* still exhibits advantages in Chinese STR. Meanwhile, similar accuracy variants are also observed when *NRTR + DPTR v.s.* NRTR, and *ABINet + DPTR v.s.* ABINet. These results demonstrate the effectiveness of DPTR in Chinese recognition.

**Multi-language Mixed Dataset.** Similarly, a multi-language mixed DPTR is trained using Multilingual CLIP and tested on MLT17. The results are given in Tab. 8 (each language is abbreviated using its first three characters). Compared with the raw implementation of NRTR, ABINet, and PARSeq, *NRTR + DPTR*, *ABINet + DPTR*, and *PARSeq + DPTR* gain accuracy improvements of 1.9%, 1.1%, and 1.6%, respectively. Note that *NRTR + DPTR* and *PARSeq + DPTR* report improvements on all the evaluated languages, while *ABINet + DPTR* reports slightly lower accuracy on Ara and Kor. This is because ABINet relies on external language models, which are not readily available for minority languages thus the side affection may be more apparent. Nevertheless, the experiment validates that DPTR can still take effect for a recognition

**Table 8: Comparison on MLT17.**

| Method | Ara | Ban | Chi | Jap | Kor | Lat | Avg |
|---|---|---|---|---|---|---|---|
| NRTR[50] | 96.1 | 94.2 | 90.4 | 90.5 | 95.1 | 95.9 | 93.7 |
| ABINet[21] | 95.5 | 93.2 | 86.2 | 89.1 | 95.9 | 94.4 | 92.4 |
| PARSeq[5] | 96.2 | 93.3 | 91.1 | 91.0 | 95.0 | 96.0 | 93.8 |
| NRTR+DPTR | **97.5** | 95.8 | **91.9** | 93.5 | **96.8** | **97.8** | **95.6** |
| ABINet+DPTR | 94.8 | 93.7 | 90.2 | 91.1 | 95.1 | 96.0 | 93.5 |
| PARSeq+DPTR | **97.5** | **95.9** | 91.5 | **93.7** | 96.0 | 97.6 | 95.4 |

task involving 10 languages without the need for language-specific preprocessing, demonstrating its great cross-language applicability.

## 5 CONCLUSION

In this study, we have presented DPTR, a novel decoder pre-training approach for STR. We have observed that embeddings extracted from the CLIP text encoder are more similar to embeddings of real text images rather than commonly employed synthetic text images. Therefore, DPTR is featured by leveraging CLIP text embeddings to pre-train the decoder, offering a new paradigm for STR pre-training. We have developed ORP, a dedicated data augmentation to generate rich and diverse embeddings and make our decoder pre-training effective, and FMU to condense the embeddings of real text images and make them better aligned with the pre-trained decoder. Extensive experiments across various decoders and languages demonstrate the effectiveness of DPTR. Our exploration above basically validates that CLIP can be utilized to enhance the training of STR models. In future, we plan to investigate the more thorough utilization of large pre-trained models like CLIP, and activate the rich knowledge contained by them to further improve the accuracy of STR as well as other OCR-related tasks.

## ACKNOWLEDGMENTS

This work was supported by National Natural Science Foundation of China (No. 32341012, 62172103). The computations in this research were performed using the CFFF platform of Fudan University.

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
