# OpenReview forum: "Decoder Pretraining with only Text for Scene Text Recognition"
_acmmm.org/ACMMM/2024/Conference — MM2024 Poster_

### Official Review · Reviewer_vHZW · 2024-05-23

**Rating:** 5
**Confidence:** 3

**Summary:**

This paper proposes a novel method (DPTR) for pre-training the decoder of text recognition models relying only on text. DPTR employs the embedding generated by CLIP's text encoder as the visual embedding of the scene text, which is fed into the decoder.  In order to enhance the diversity of the embedding produced by the text encoder, the Offline Randomized Perturbation (ORP) strategy is proposed, which incorporates the embedding of images produced by the image encoder of CLIP. DPTR pre-training of the existing sota method shows a major performance improvement.

**Strengths:**

1. The proposed DPTR method for decoder pretraining with only text is overall insightful. DPTR exploits CLIP's alignment of visual and text, and thus simulates visual embedding using a text encoder.
2. The experiments are comprehensive, including performance in English, Chinese, and multilingualism, as well as detailed experiments on the ablation of individual modules.

**Limitations:**

1. The proposed FMU module is a cross attention module, I can't quite figure out why it's called a Feature Merge Unit as it doesn't seem to have a feature fusion function.
2.  What are the advantages of FMU over other approaches such as linear, mlp, convolution layers? Comparative experiments are missing.
3.  Missing references to Scene text recognition related articles in CVPR 2024. Such as  "OTE: Exploring Accurate Scene Text Recognition Using One Token", "Multi-modal In-Context Learning Makes an Ego-evolving Scene Text Recognizer" and "An Empirical Study of Scaling Law for OCR ".

**Suitability:**

2

---

### Official Review · Reviewer_8fsV · 2024-05-23

**Rating:** 3
**Confidence:** 3

**Summary:**

This paper focuses on the pre-training of scene text recognition and proposes a pre-training method for the decoder where the decoder tries to predict the word based on the text embedding from the CLIP. Furthermore, the Offline Random Perturbation and Feature Merge Unit are designed to improve the robustness of the model.

**Strengths:**

1. Pre-training the decoder is novel for the scene text recognition;
2. Using the text embedding from the CLIP is neat since the visual and semantic features are aligned in the CLIP.

**Limitations:**

1. More details about the usage of F_p can be provided, will it work as the key and value of the attention mechanism?
2. The proposed method seems can not be applied to the CTC-based decoder;
3. I wonder if initializing the image encoder with the CLIP can achieve a better performance;
4. The compared methods in Table 6 is not up-to-date;
5. Since the paper focuses on the pre-training of text recognition, treating the FMU as a contribution seems not appropriate, since it only be adopted for fine-tuning;
6. Lacks a comparison between the proposed method with other unsupervised pre-training of STR.

**Suitability:**

2

---

### Official Review · Reviewer_u4VE · 2024-05-24

**Rating:** 4
**Confidence:** 4

**Summary:**

The paper introduces a novel pre-training method named DPTR (Decoder Pretraining with only text for STR) for scene text recognition (STR). Unlike traditional methods that rely on synthetic datasets or real images, DPTR leverages text embeddings produced by the CLIP text encoder as visual embeddings to train the decoder. The proposed method includes an Offline Randomized Perturbation (ORP) strategy, which enriches text embeddings with image embeddings to prevent overfitting. Additionally, a Feature Merge Unit (FMU) is introduced during fine-tuning to enhance the decoder's focus on character regions within text images. The experimental results demonstrate that DPTR achieves state-of-the-art performance across various decoders and multi-language STR tasks.

**Strengths:**

1. The paper proposes an interesting approach to STR pre-training by using text embeddings from CLIP, eliminating the need for text images during pre-training. This is a significant departure from traditional methods that rely heavily on synthetic or real text images. By using text embeddings as visual embeddings, the method effectively addresses the domain gap between synthetic and real images

2. The introduction of the Offline Randomized Perturbation (ORP) strategy helps in enriching text embeddings and prevents the model from overfitting, which enhances the robustness of the pre-training process. The Feature Merge Unit (FMU) helps focus the model’s attention on the character regions during fine-tuning, thereby improving accuracy and efficiency.

3. The experiments and ablations in this paper are detailed and comprehensive, showing improved performance and achieving state-of-the-art results.

**Limitations:**

1. With the innovative training approach, the design of the model structure is relatively simple.

2. Some additional evaluation should be included. Since the CLIP model is trained on large-scale image-text pair datasets, the proposed method in this paper may not be robust to noise. The authors can compare the performance on occlude datasets such as HOST, WOST, and WordArt to enhance persuasiveness.

3. Discussion about the computation complexity of the proposed ORP and FMU should be included.

**Suitability:**

3

---

### Official Review · Reviewer_XEFu · 2024-05-27

**Rating:** 2
**Confidence:** 3

**Summary:**

This paper introduces Decoder Pretraining with Only Text for Scene Text Recognition (DPTR), leveraging CLIP's text embeddings to bridge the gap between synthetic and real images. DPTR uses text embeddings as visual proxies during pre-training and incorporates an Offline Randomized Perturbation (ORP) strategy to enhance these embeddings with image information. In the fine-tuning stage, a Feature Merge Unit (FMU) focuses on character regions, resulting in improved accuracy and efficiency, with extensive experiments demonstrating DPTR's superior performance across various decoders and languages.

**Strengths:**

The paper showed some improvement compared with baselines.

**Limitations:**

1. The improvement in performance compared to the baselines is relatively modest.

2. The novelty of the approach appears limited. Previous works such as ABINet and SEED have already utilized external pretrained language models to leverage linguistic priors, which are trained with text-only data and enhance scene text recognition models. Pretraining the decoder with text, in this context, seems like an incremental advancement.

**Suitability:**

2

---

### Meta-Review · Area_Chair_aEw2 · 2024-07-01

**Recommendation:** Accept (Poster)
**Confidence:** 5

**Metareview:**

The authors did a good rebuttal. The reviewers unanimously recommend acceptance in the final rating. After checking the rebuttal, the review, and the paper, the AC agrees with this assessment.